# Mitochondrial DNA is a sensitive surrogate and oxidative stress target in oral cancer cells

**Jingyu Tan[1], Xinlin Dong[1], Haiwen Liu[1,2]***

**1** The First Affiliated Hospital of Jinzhou Medical University, Jinzhou, China, **2** Liaoning Provincial Key Laboratory of Clinical Oncology Metabonomic, Jinzhou, China

* haiwen163@163.com

**Data Availability Statement:** The data generated in this study is included in the paper itself and uploaded as supplementary information.

## Abstract

Cellular oxidative stress mediated by intrinsic and/or extrinsic reactive oxygen species (ROS) is associated with disease pathogenesis. Oxidative DNA damage can naturally be substituted by mitochondrial DNA (mtDNA), leading to base lesion/strand break formation, copy number changes, and mutations. In this study, we devised a single test for the sensitive quantification of acute mtDNA damage, repair, and copy number changes using super-coiling-sensitive quantitative PCR (ss-qPCR) and examined how oxidative stress-related mtDNA damage responses occur in oral cancer cells. We observed that exogenous hydrogen peroxide ($H_2O_2$) induced dynamic mtDNA damage responses, as reflected by early structural DNA damage, followed by DNA repair if damage did not exceed a particular threshold. However, high oxidative stress levels induced persistent mtDNA damage and caused a 5–30-fold depletion in mtDNA copy numbers over late responses. This dramatic depletion was associated with significant growth arrest and apoptosis, suggesting persistent functional consequences. Moreover, oral cancer cells responded differentially to oxidative injury when compared with normal cells, and different ROS species triggered different biological consequences under stress conditions. In conclusion, we developed a new method for the sensitive detection of mtDNA damage and copy number changes, with exogenous $H_2O_2$ inducing dynamic mtDNA damage responses associated with functional changes in stressed cancer cells. Finally, our method can help characterize oxidative DNA damage in cancer and other human diseases.

## Introduction

Oral squamous cell carcinoma (OSCC) is a cancer with high morbidity and mortality [1]. In 2018, combined effects of lip, oral cavity, and oropharynx cancers resulted in an estimated 228,389 fatalities and 447,751 new cancer cases, or 2.4% of all cancer deaths worldwide [2]. Oral cancer is a serious public health issue, particularly for dentists. The disease is among the top ten cancers in terms of incidence, while survival rates have not greatly increased recently, despite advancements in treatment and research, posing a persistent challenge to biomedical science [3]. In around 90% of cases, alcohol and smoking are key risk factors for oral cancer, which is a preventable disease [4] and they exert synergic effects [5]. Intrinsic oxidative stress

**Funding:** The Science Foundation of the Liaoning Provincial Department of Education (LJKMZ20221224),the Science and Technology Program Project of the Liaoning Province(2023JH/101700235),the funding of Scientific Research of The First Affiliated Hospital of Jinzhou Medical University(FYQKR-202203).

**Competing interests:** The authors have declared that no competing interests exist.

**Abbreviations:** Abbreviation, Full Name; $H_2O_2$, Hydrogen peroxide; ROS, reactive oxygen species; ss-qPCR, Supercoiling-sensitive PCR; DMEM, Dulbecco's Modified Eagle's Medium; DMSO, Dimethyl sulfoxide; ETO, Etoposide; MTT, 3-(45-dimethylthiazol-2-yl)-25-diphenyltetrazilium; FCM, Flow cytometry.

due to augmented cellular reactive oxygen species (ROS) production is increasingly intrinsic to many cancers [6]. ROS comprise a family of short-lived molecules, such as superoxide anion($O^2$.), hydrogen peroxide ($H_2O_2$) and hydroxyl radicals (OH),first described as free radicals in skeletal muscle [7]. They are extremely reactive molecules that contain oxygen and have the ability to damage DNA and affect damage responses [8].

In organisms, ROS is constantly generated during normal aerobic metabolism. Low ROS doses, particularly $H_2O_2$, are mitogenic and promote cell proliferation and induce mutagenesis, with high levels not only inhibiting cell proliferation, but also inducing elevated cytotoxicity in cells and causing apoptosis in several tumor types [9–12]. Of the different ROS types, $H_2O_2$ is a hallmark molecule generated by almost all oxidative stress sources, and is a vital oxygen metabolite that plays significant roles in diseases driven by oxidative stress, such as inflammation [13, 14].

Physical or chemical alterations to DNA can indicate DNA damage, which impacts the interpretation and transfer of genetic information. Numerous external and internal insults, such as chemicals, radiation, free radicals, and MicroRNAs(miRNAs), maladjusted in multifarious malignant tumor, can be considered as both carcinogens and tumor-inhibiting factor [15], can harm DNA, and each one does so in a different way.

The main ROS molecules formed by mitochondria are superoxides ($O^{2-}$) and $H_2O_2$ [16]. Cellular DNA and other macromolecules are directly oxidatively damaged by HO•, the most reactive kind of ROS [17]. Some features of cancer, including transcription factors and activated proto-oncogenes, genomic instability, resistance to treatment, invasion, and metastasis, may be partially explained by persistent oxidative stress [18]. The mitochondrial electron transport chain, is one source of cellular ROS, which is generated as by-products. Mitochondrial DNA (mtDNA), which is circular and multi-copy, is susceptible to oxidative DNA damage. Its distinct characteristics include a high copy number, susceptibility to damage, and dependence on repair processes [6, 19].

In this study, we developed a new method for the sensitive detection of mtDNA damage, copy number changes, and exogenous $H_2O_2$ production induced by dynamic mtDNA damage responses associated with functional changes in oral cancer cells. We may be able to evaluate oxidative DNA damage in cancer and other disorders using our sensitive mtDNA test.

## Materials and methods

### Reagents and cell culture

Normal human fibroblast (BJ) and human oral squamous carcinoma (SCC-25) cell lines were obtained from the American Type Culture Collection (Manassas, VA, USA). We bought the majority of the chemicals from Sigma-Aldrich in Oakville, Ontario, Canada. Whereas SCC-25s were maintained in DMEM/F12 along with 15 mM HEPES, 1.2 g/L sodium bicarbonate, 0.5 mM sodium pyruvate, and 400 ng/ml hydrocortisone, BJ cells were cultured in Dulbecco's Modified Eagle's Medium (DMEM) supplemented with 1.5 g/L sodium bicarbonate. Supplements of 10% fetal bovine serum, 100 μg/mL streptomycin, and 100 U/mL penicillin were added to both mediums. Cells were cultivated at 37°C in a humidified 5% $CO_2$ environment.

### Inducing oxidative DNA damage

SCC-25 and BJ cells ($1\times10^6$ cells) were grown for 24 h. A 1 M $H_2O_2$ (working solution) was freshly prepared in phosphate buffered saline (PBS). For exposure studies, culture dishes containing cells were treated with different $H_2O_2$ concentrations in serum-free medium for 15 min and 60 min. For recovery studies, $H_2O_2$ was applied for 60 min and cells allowed recover

for 48 h in fresh complete medium. Cells were washed once in PBS, collected by trypsin digestion, and genomic DNA extracted.

## DNA preparation

Using QIAGEN Blood and Cell Culture DNA kits (Qiagen, Germany), total DNA from cell pellets was extracted in accordance with the manufacturer's instructions with a few minor adjustments to preserve mtDNA [20]. DNA was quantified using a NanoDrop (Thermo Scientific, Amarican); $1 \times$ Tris/EDTA (pH 8.0) was used to prepare a 1 ng/L template solution. Template solutions were divided equally into two parts: half the original template was used to quantify relaxed/damaged mtDNA, and the other half underwent heat treatment for six minutes at 95°C to measure the amount of whole mtDNA.

## Real-time PCR with supercoiling sensitivity for quantifying mtDNA damage

The total mtDNA copy number and mtDNA damage level calculation method was previously reported [21, 22]. Primer sequences are shown (Table 1). The new two-step procedure for determining mtDNA damage is described: 38 cycles of a two-step reaction at 80.0°C for 6 s and 61.0°C for 30 s are followed by 95.0°C for 30 s, 61.0°C for 30 s, 95.0°C for 3 s, and 61.0°C for 30 s. Melt curve analysis was performed at amplification end. We used the ViiA7[TM] Real-Time PCR System with Power SYBR®Fast Green PCR Master Mix (ABI, American). The percentage of relaxed mtDNA fraction in samples compared to total copy numbers was used to calculate mtDNA structural damage, while relative mtDNA copy numbers were normalized to total copy numbers in controls.

## 3-(4,5-dimethylthiazol-2-yl)-2,5-diphenyltetrazolium bromide (MTT) assays

These tests were based on the process by which mitochondrial reductase changes the water-soluble yellow dye MTT into the insoluble purple formazan [23]. Different functional end-points were examined when varying $H_2O_2$ concentrations were employed for one, twenty-four, and forty-eight hours. The long-term MTT impact suggests cell viability and/or growth inhibition, whereas the 56short-term MTT effect is dictated by mitochondrial respiration and cellular redox status effects.$6x10^3$ cells (150 uL/well) were cultured in 96-well plates for a full night at 37°C with 5% $CO_2$ until the cells adhered. The next day, growth media was created with various $H_2O_2$ concentrations (during 24 and 48 hours of incubation). A negative control was established using untreated material. Following a 24-hour incubation period, wells were treated with 5 mg/mL MTT (Sigma, UK) for 4 hours at 37°C in 5% $CO_2$, and then formazan was solubilized with the addition of dimethyl sulfoxide. Using an ELx808TM Absorbance Microplate Reader (BioTek, USA), plates were analyzed at 550 nm.

**Table 1. Primer sequences for long PCR and real time PCR amplifications.**

| Primers | Forward 5'-3' | Reverse 5'-3' |
|---|---|---|
| CO2(3285bp) | CCTAGGGTTTATCGTGTGAG | CTAGTTAATTGGAAGTTAACGG□ |
| Calicin(2658bp) | ATTCCAGAAGCCTTTAACTAG | ACAAATGAGACACAAACTACCG |
| CO2(real-timePCR) | CCCCACATTAGGCTTAAAAACAGAT | TATACCCCCGGTCGTGTAGCGGT |
| Calicin(real-timePCR) | CTGGTCGCTACATCTACATCTC | CAGGTCAGGCAACTTGGTC |

## Quantifying $\gamma$-H$_2$AX in DNA double-strand breaks (DSBs) by flow cytometry

Assessing the amounts of γ-H2AX offers a dependable and sensitive way to measure responses to DNA damage in many cell types [24, 25]. In short, the cells were in blocking buffer (5.0% rabbit serum (Labtech) + 0.1% Triton TM X-100 (Sigma-Aldrich, Dorset, UK) in PBS) and gently shaken for one hour at room temperature. Blocking buffer was removed, and cells were then gently stirred at 4˚C for 24 hours before being treated with a primary antibody (anti-phosphohistone γ-H2AX (serine 139) mouse monoclonal IgG1 antibody; clone JBW301 (Millipore, Waterford, Ireland) was diluted 1/10000 in blocking buffer). The main antibody was eliminated after two PBS washes with 0.1% Triton TM X-100. To prepare cells for flow cytometry, they were resuspended in PBS. Information on 5000–10000 cells was gathered.

## Alkaline gel electrophoresis to detect H$_2$O$_2$-induced DSB and single-strand breaks (SSBs)

Alkaline gel electrophoresis is performed under alkaline conditions to promote DNA denaturation to better detect SSBs [26, 27]. Approximately 3–5 μL DNA was added to 3.2 μL of a 500 mM NaOH stock solution. The final NaOH concentration was adjusted to 100 mM (50 mM HEPES buffer). Then, 2 μL loading buffer was added and incubated with DNA at 37˚C for 20 min. After loading samples onto the gel, buffer was circulated around the system. For sixteen hours, electrophoresis was run at 30 V. The next day, the gel was placed into neutralizing solution for 20 min at room temperature, then rinsed twice in deionized water, and placed into a container with enough 10000 × Gel stain for 40 min at room temperature. The waste agarose gel was disposed of in the designated rubbish bin and disposed of centrally.

## Data analysis

Statistical analyses were performed in GraphPad Prism version 5 (GraphPad, San Diego, CA, USA). One-way analysis of variance with Dunnet's multiple comparison tests were used for statistical analyses involving > two groups. Otherwise, Student's t-tests were used to analyze specific samples/groupings.

## Results

**Ss-qPCR sensitivity and stability in detecting mtDNA base damage.** Using ss-qPCR, we developed a sensitive quantification method for acute mtDNA damage, repair and copy number changes in a single test. The two-phase protocol allows for the sensitive identification of basal mtDNA damage levels resulting from DNA strand breaks and the estimation of mtDNA content while also greatly reducing experimental artifacts [28]. From the literature, damaged mtDNA levels in cancer cells can be as high as 30% or over and over 40% in Fast and Regular protocols [28]. Our detection results showed that in controls (no treatment), mtDNA base damage at different time points was stable, while our method identified basal mtDNA damage levels of approximately 30% of total mtDNA content in untreated SCC-25 and BJ cells (Fig 1), which was a significant reduction. Studies were repeated three times or more.

**H$_2$O$_2$ induces prevalent mtDNA damage in oral cancer and normal cells.** Using our approach, mtDNA damage was assessed to investigate if oxidative DNA damage linked with differential cell toxicity between oral cancer and normal cells caused by exogenous H$_2$O$_2$. When SCC-25 cells were treated with 120–960 μM H$_2$O$_2$, mtDNA damage increased during exposure durations of 15 and 60 minutes in a dose-dependent manner (Fig 2A). When SCC-25 cells, previously treated with 480 μM H$_2$O$_2$ and allowed recover in complete medium, mtDNA damage maintained over time and displayed modest healing at 24 h, as shown by

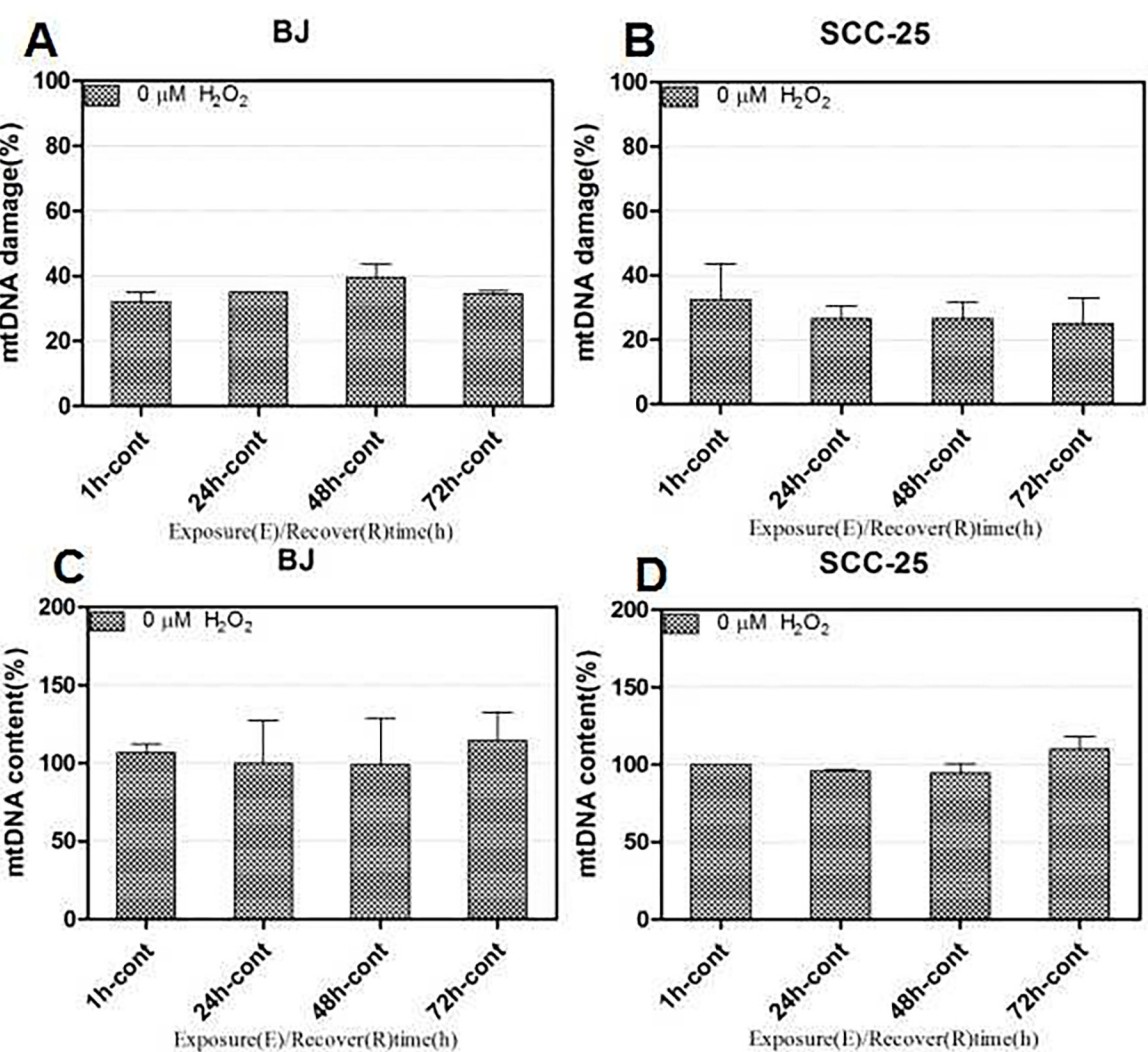

**Fig 1. Two-phase, supercoiling-sensitive qPCR for improved mtDNA damage detection.** Damaged mtDNA percentages in SCC-25 oral cancer and BJ cell lines (A–D). Data were calculated using Image-Pro 5.0 software (Media Cybernetics).

reduced damage molecule ratios(Fig 2B). In 60 minutes, 120–240 μM $H_2O_2$ altered more than 70% of the mtDNA molecules in the cells into damaged versions, while the amount of mtDNA did not change much throughout exposure(Fig 2C and 2D). During the course of the 2-to 24-hour healing period, a 20-fold decrease in the total amount of mtDNA was also seen, indicating that the highly damaged mtDNA molecules were actively undergoing destruction (Fig 2E). We observed significant quantities of floating cells in 24 h recovery dishes. We were also surprised to observe sensitive mtDNA damage responses in BJ cells when treated with several $H_2O_2$ doses, e.g., at 240 μM $H_2O_2$, considerable mtDNA damage was seen, and for one hour, a dose-dependent increase was brought on by 240–960 μM $H_2O_2$ (Fig 2F). Moreover, 480 μM $H_2O_2$ induced > 60% early structural damage in BJ cells (Fig 2G). BJ cells treated with 960 μM $H_2O_2$ showed much more extensive mtDNA copy number loss (Fig 2H). Therefore, mtDNA damage and destruction caused by $H_2O_2$ was common in both normal and oral cancer cells, but it was not associated with distinct cell cytotoxicity.

**Oral cancer and normal cells exhibit selective cytotoxicity from exogenous $H_2O_2$.** We used MTT assays to assess oral cancer's cellular oxidative damage vs. normal BJ cells.

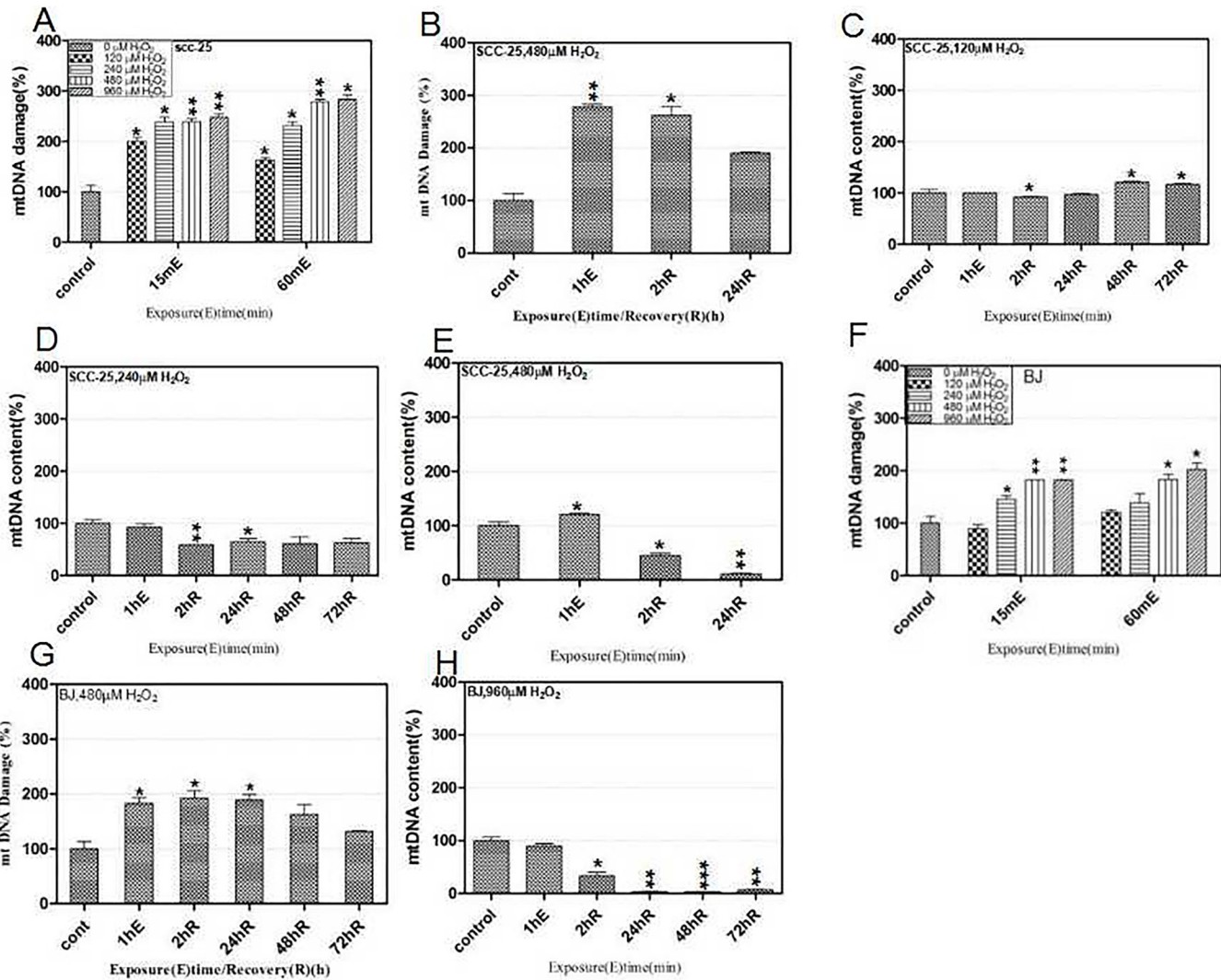

**Fig 2. Hydrogen peroxide ($H_2O_2$) induces prevalent mtDNA damage in oral cancer and normal cells.** $H_2O_2$-induced mtDNA damage, repair, and copy number depletion were analyzed using ss-qPCR. SCC-25 cells were treated for 15 and 60 min with 120–980 µM $H_2O_2$ to assess dose-responses during mtDNA damage (A). To measure repair activity, cells were treated for 60 min with 480 µM $H_2O_2$ and then allowed recover for 2 h and 24 h (B). SCC-25 cells were treated for 60 min with 120–480 µM $H_2O_2$ and then allowed recover for 2 h and 72 h to measure copy number changes (C, D, E). BJ cells were treated for 15 and 60 min with 120–980 µM $H_2O_2$ to assess dose-responses during mtDNA damage (F). BJ cells were treated for 60 min with 480 µM $H_2O_2$ and then allowed recover for 2–72 h to assess copy number changes (G). BJ cells were treated for 60 min with 960 µM $H_2O_2$ and then allowed recover for 2–72 h to measure copy number changes (H).

Comprehensive $H_2O_2$-induced dose- and time-dependent profile responses were generated in cells in preliminary analyses. At one hour, both cell lines showed much higher vulnerability to early $H_2O_2$ toxicity(Fig 3A and 3B) and 24 hours later, growth inhibition (Fig 3C). In SCC-25 cells, at 24 hours, 287.2 µM of $H_2O_2$ was needed to achieve 50% growth inhibition (EC50), whereas in contrast, BJ cells demonstrated robust resilience against external oxidative damage, demonstrating an EC50 value of 578.4 µM $H_2O_2$(Fig 3D).

## $H_2O_2$ and etoposide (ETO)-induce DNA SSBs and DSBs in oral cancer cells

We used flow cytometry to investigate the effects of $H_2O_2$-induced DNA breaks in oral cancer cells; we used 480 µM $H_2O_2$ at 1 h/2 h/6 h, and 24 h and the γ-$H_2AX$ antibody to detect DNA

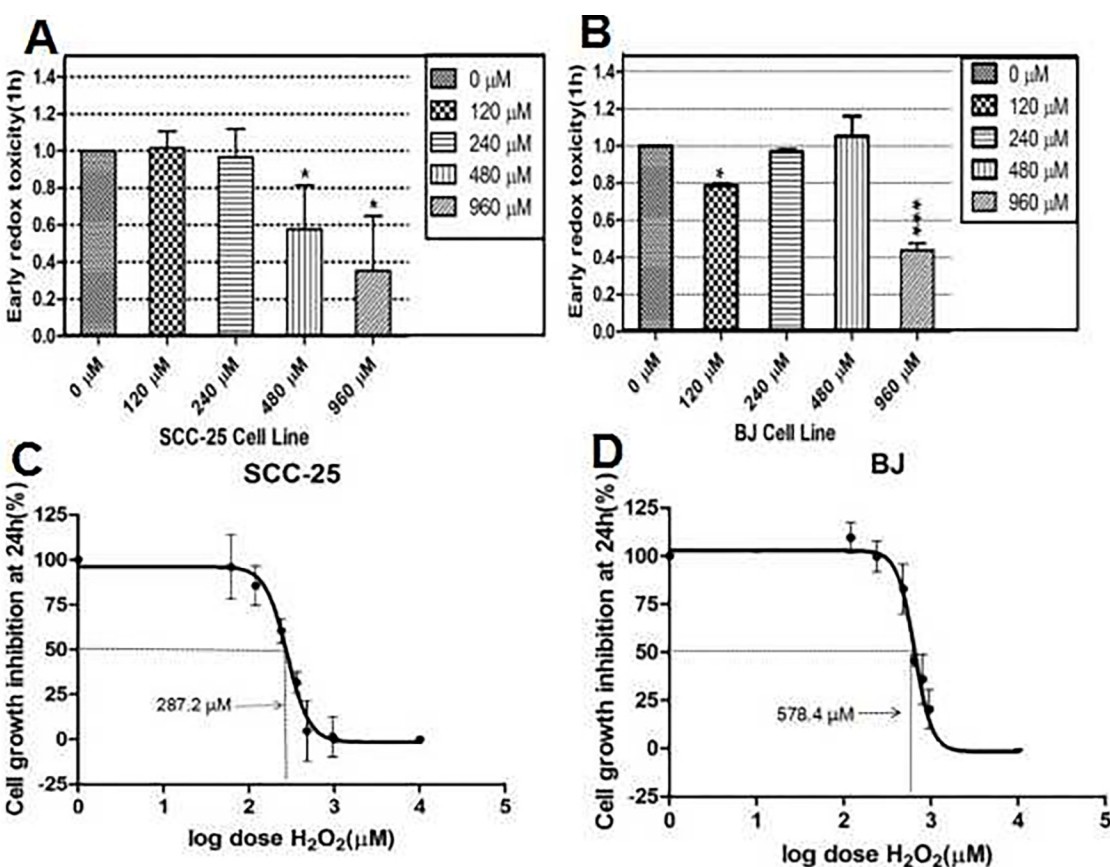

**Fig 3. Hydrogen peroxide ($H_2O_2$) induces differential cell toxicity in oral cancer and normal cells.** $H_2O_2$ induces early redox toxicity at 1 h; expressed as the percentage of treatment vs. control cells (A, B), and growth inhibition at 24 h; expressed as a function of log dose (C, D) in MTT assays in cell lines. SCC-25 cells were treated with final 0/120/240/480, and 960 μM $H_2O_2$ concentrations. BJ cells were exposed to 0/120/240/480 and 960 μM $H_2O_2$ concentrations at indicated times. 50% growth inhibition (EC50) is indicated by a solid line. Statistical significance is indicated by: $p < 0.05$ (*), $p < 0.01$ (**), and $p < 0.001$ (***). Data were calculated using Image-Pro 5.0 software (Media Cybernetics). RFI/cell mean values were obtained from at least two separate experiments.

fracture points of 4.14%, 3.23%, 3.66%, and 3.58%, respectively (Fig 4A–4D). The break point in the control group was 0.68%. Although experimental group data were statistically significantly different when compared with controls, a dose relationship was not observed at different stimulation times. We also used another chemical reagent ETO, which is a semisynthetic derivative of podophyllotoxin and is frequently used to treat solid tumors, lymphoma, and leukemia, among other cancers [29]. ETO is a potent DNA DSB inducer via topoisomerase II inhibition. The medication is utilized as an efficient chemotherapy [30].

Gamma-$H_2AX$ is a key factor during damaged DNA repair processes; it is recruited to damage sites where it recruits other DNA repair machinery [31, 32]. Phosphorylated γ-$H_2AX$ status at Ser139 ($H_2AX$) was examined after cells were treated with 50 μg/mL ETO for 1 h/2 h /6 h, and 24 h. We identified 16%/22.1%/35.1%, and 65.5% breaks, respectively. Gamma-H2AX production is a sensitive and fast cellular response to double-strand breaks (DSBs) that may reveal information about higher order chromatin structures; utilizing the ss-qPCR method, exogenous $H_2O_2$ causes widespread and sensitive mtDNA damage responses in both normal and oral cancer cell lines.

**$H_2O_2$-induces DNA SSBs in oral cancer cells.** Agarose gel electrophoresis is used for DNA damage analyses [33]. Nuclear DNA damage was determined using γ-$H_2AX$ antibodies

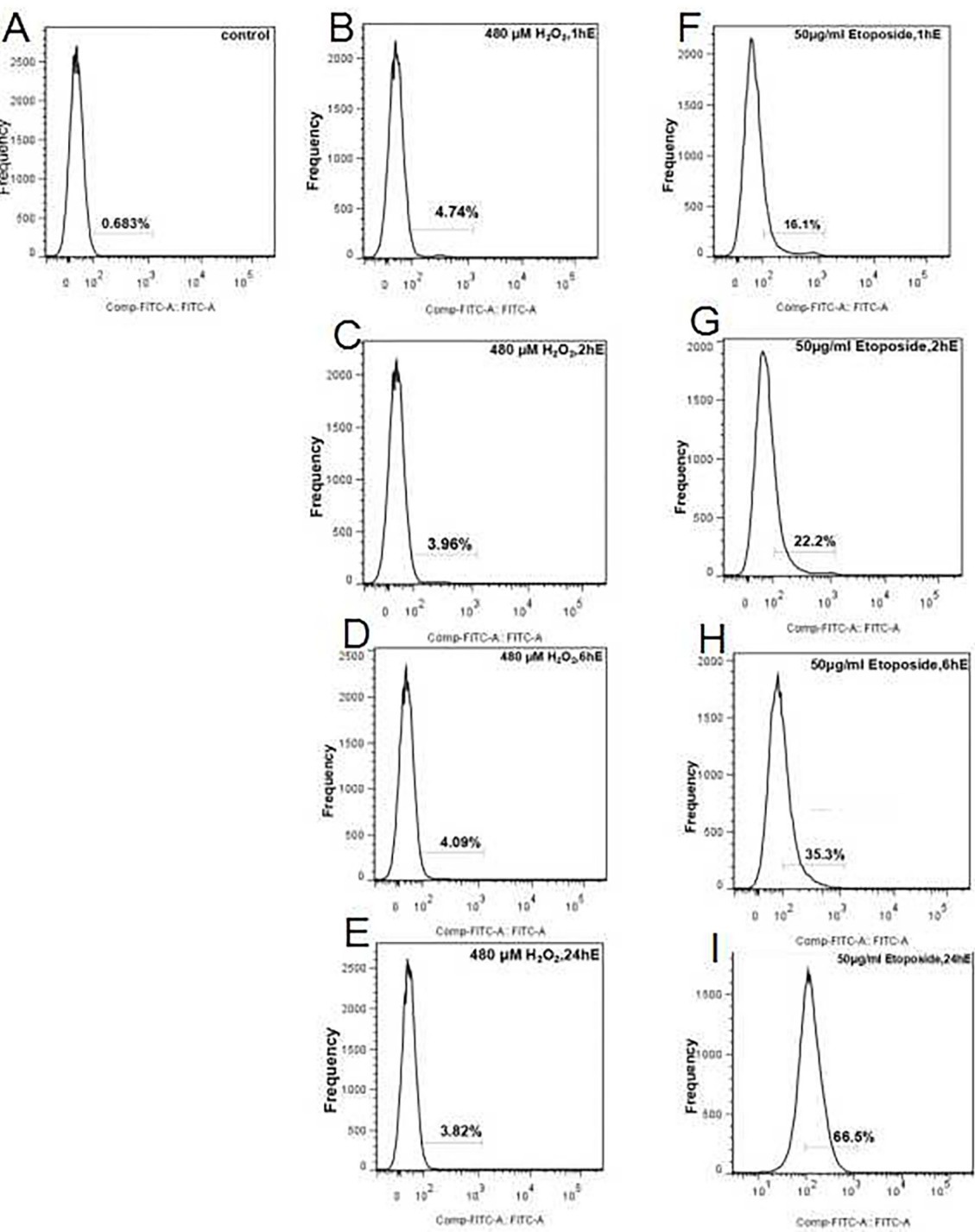

**Fig 4. Hydrogen peroxide (H₂O₂)- and etoposide (ETO)-induce DNA breaks in SCC-25 cells. 480 μM H₂O₂, 50 ng/L ETO two chemicals stimulated 1 h/2 h/6 h/24 h, respectively, using flow cell surgery for nuclear DNA fracture detection.** (A) Blank control group; (B–E) 480 μM H₂O₂ stimulation at 1 h /2 h /6 h /24 h, nuclear DNA fracture ratio; (F–I) 50 ng/L ETO stimulation 1 h/ 2 h/ 6 h/ 24 h, nuclear DNA fracture ratio.

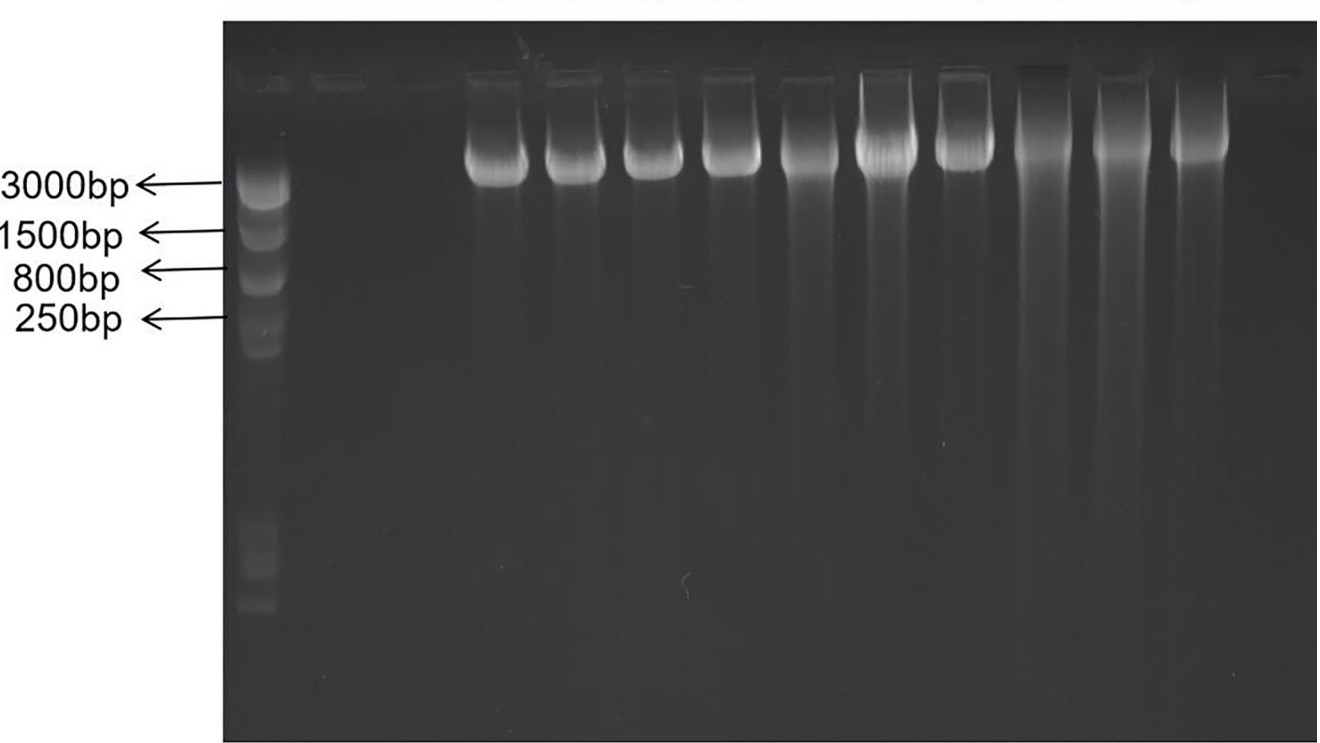

**Fig 5. Single Stranded Break (SSB) detection in SCC-25 cells continuously exposed to 120/240/480 μM hydrogen peroxide (H₂O₂) for 1 h/2 h/24 h.**
LaneA—control; Lane B—120 μM H₂O₂ exposure for 1 h; Lane C—120 μM H₂O₂ exposure for 2 h; Lane D—120 μM H₂O₂ exposure for 24 h; Lane E—240 μM H₂O₂ exposure for 1 h; Lane F- 240 μM H₂O₂ exposure for 2 h; Lane G—240 μM H₂O₂ exposure for 24 h; Lane H—480 μM H₂O₂ exposure for 1 h; Lane I—480 μM H₂O₂ exposure for 2 h; and Lane J—480 μM H₂O₂ exposure for 24 h.

combined with flow cytometry and alkaline gel electrophoresis. SSBs were observed in SCC-25 cells when continuously exposed to 120/240/480 μM H₂O₂ for 1 h/2 h/24 h. SSBs were visualized in cells using OTX-coupled AGE analysis. Images were obtained using Gei stain and a Kodak Image Station 440CF system. When cells were stimulated for 1 h at 120/240/480 μM concentrations, the degree of nuclear DNA damage increased with increased concentrations (Fig 5, lines B, E, H). At 2 h stimulation, the degree of nuclear DNA damage was dose-dependent (Fig 5, lines C, F, I), but differences were not significantly different to the 1 h stimulation. At 24 h stimulation, the degree of nuclear DNA damage increased over time, but some damage was recovered (Fig 5, Lines D, G, J).And the comparison of the intensity of DNA fluorescence, showed that the damage trend of nuclear DNA was the same as mt DNA(Fig 6)

## Discussion

In this study, dynamic changes during mtDNA damage, repair, and copy number in oral squamous carcinoma cells were induced by exogenous oxidative stimulation. We showed that when low oxidative stimulation levels were applied to cells, mtDNA exhibited some damage responses in the short term, then the body's repair mechanisms soon took effect and quickly returned mtDNA to normal levels, this process reflects the dynamic changes in mtDNA damage and repair in oral squamous cell carcinoma cells. MtDNA damage and repair in cells was dynamically altered, while copy numbers did not increase or decrease due to repair mechanisms. Copy numbers were also significantly reduced by approximately 20-fold when

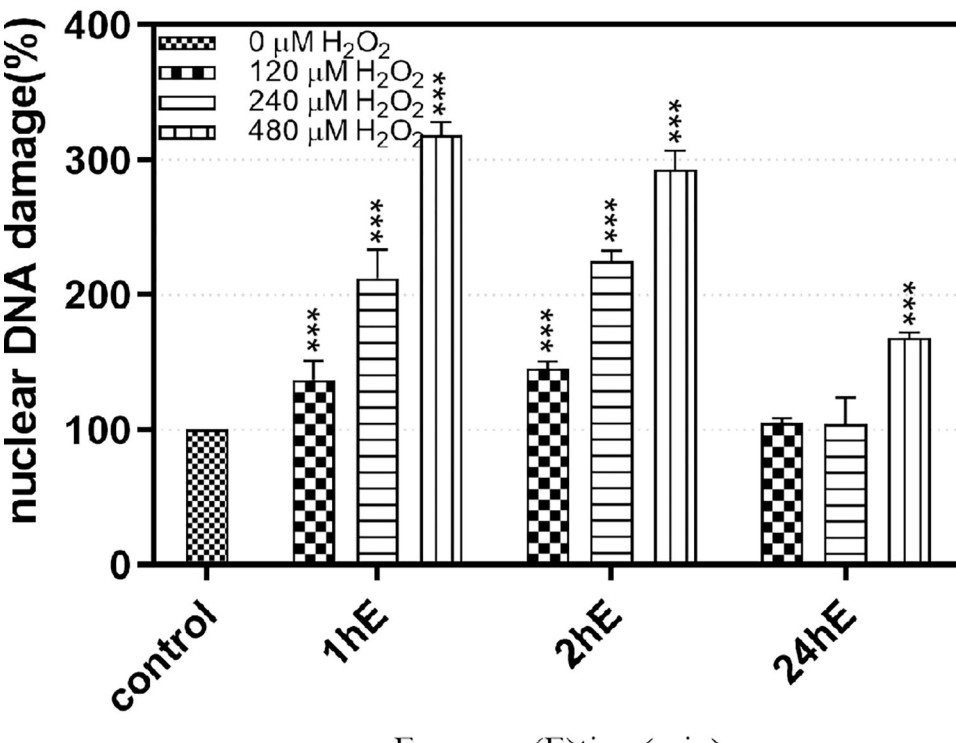

**Fig 6. SSB detection in SCC-25 cells exposed continuously to 120/240/480μM H$_2$O$_2$ for 1/2/24 h.** The picture shows the comparison of the intensity of DNA fluorescence.

compared with controls. Under similar stimulation conditions, control cells showed changes in oxidative damage only at high concentrations, and decrease copy numbers at 2 h and 24 h, which quickly reverted to normal levels. Only at very high concentrations did copy numbers significantly decrease. Thus, oral squamous carcinoma cells were more sensitive to oxidative stimulation.

In the literature, it was suggested that changes in gene copy numbers are important factors affecting gene function, and that some tumors are not triggered by mutations but by increases/ decreases in gene copy numbers. Vijay *et al.* [34] used *in situ* digoxigenin-labelled mtDNA probes to detect mtDNA in malignant and benign cells and identified significant increases in mtDNA copy numbers in malignant cells. Boultwood *et al.* [35] used *in situ* hybridization probes to detect mtDNA in peripheral blood or bone marrow cells from leukemia patients and normal subjects, and reported significantly increased mtDNA copy numbers in all acute leukemia and most chronic leukemia patients (8/9). Increased mtDNA copy numbers may be related to under- developed mtDNA repair mechanisms and inefficient repair, while mutations in mtDNA cause functional defects in mitochondria, the production of which requires excessive mtDNA replication to compensate for functional defects. It was also reported that differences in mtDNA copy numbers existed between gastric cancer and normal tissue, with significant decreases in copy numbers in gastric cancer cells [34]. This may be due to ROS damage to mtDNA or D-loop regions, thereby causing prolonged mtDNA replication cycles, reduced replication or increased mtDNA damage. Thus, genetic structure changes in tumor

mtDNA, such as point mutations in non-coding regions or microsatellite instability, may affect regulatory sequences in non-coding regions, thereby altering mtDNA transcription and replication regulator or inducer infinity to D-loops and altering mtDNA copy numbers [35–38].

To assess nuclear DNA breaks in SCC-25 cells after $H_2O_2$ stimulation, flow cytometry, γ-$H_2AX$, and alkaline gel electrophoresis assays were performed. We observed significant difference between $H_2O_2$-stimulated samples when compared with controls, but difference were not significant. However, different stimulation times showed certain dose-dependent relationships. When we used ETO, a chemical agent which induced DNA DSBs, DNA break points were not only significantly different to controls, but dose differences were identified depending on time differences. It was clear from both chemical stimuli (ETO and $H_2O_2$) that while both reagents caused severe DNA breaks, the γ-$H_2AX$ method was only sensitive to DSBs and could not detect SSBs, the $H_2O_2$ can cause DNA SSBs. The γ-$H_2AX$ assay is applicable to DSB detection, whereas our ss-qPCR assay was more sensitive and stable with respect to mtDNA oxidative damage.

Our alkaline gel electrophoresis studies, which detected a full range of DNA breaks, confirmed that oxidative stimulation conditions triggered nuclear DNA damage in a dose-dependent manner. Taken together, these results showed that the extent of nuclear DNA damage under exogenous oxidative stimulation followed the same trend as for mtDNA damage, with both indicating similar gene damage trends.

Several studies reported that when oral squamous carcinoma cells and normal skin fibroblasts are subjected to exogenous oxidative stimuli, they exhibit different degrees of damage, repair, and copy number dynamics with respect to oxidative damage. Such observations suggest that when cells are subjected to sustained oxidative stress, ROS production is increased and accumulates in tumor cells, which becomes relevant to tumorigenesis. By studying these mechanisms, researchers can prevent and treat oral squamous carcinoma. However, it remains unclear which cellular processes contribute to ROS propagation in cancer cells and how these alterations are linked to cellular oxidative damage.

## Conclusions

Using ss-qPCR, in oral cancer cells, we found dynamic responses to mtDNA damage, including sensitive early mtDNA damage, repair, and copy number changes when exogenous $H_2O_2$ was present. Increased sensitivity in cancer cells to oxidative DNA damage was consistent with preferential cell toxicity and the simultaneous induction of nuclear DNA damage. Thus, oxidative stress increases the sensitivity of oral cancer cells., and dynamic oxidative stress responses may offer fresh perspectives for the early prevention and treatment of oral cancer.

## Supporting information

**S1 Fig. This is the original data for all the data displayed in Fig 1.** The values behind the means, standard deviations and the values used to build graphs, the points extracted from images for analysis, all include in these figures.
(ZIP)

**S2 Fig. This is the original data for all the data displayed in Fig 2.** The values behind the means, standard deviations and the values used to build graphs, the points extracted from images for analysis, all include in these figures.
(ZIP)

**S3 Fig. This is the original data for all the data displayed in Fig 3.** The values behind the means, standard deviations and the values used to build graphs, the points extracted from images for analysis, all include in these figures.
(ZIP)

**S4 Fig. This is the original data for all the data displayed in Fig 4.** The values behind the means, standard deviations and the values used to build graphs, the points extracted from images for analysis, all include in these figures.
(ZIP)

**S5 Fig. This is the original data for all the data displayed in Fig 5.** The original gel electrophoresis images are included, along with the specific sample names and sizes represented by each band.
(ZIP)

**S6 Fig. This is the original data for all the data displayed in Fig 6.** The values behind the means, standard deviations and the values used to build graphs, the points extracted from images for analysis, all include in these figures.
(ZIP)

## Acknowledgments

We thank the members of our division for their contribution to this study, Jingyu Tan and Xinlin Dong are responsible for writing article, Haiwen Liu for technical assistance.

## Author Contributions

**Conceptualization:** Haiwen Liu.

**Data curation:** Xinlin Dong, Haiwen Liu.

**Writing – original draft:** Jingyu Tan.

**Writing – review & editing:** Haiwen Liu.

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
