## [Decision Letter · Decision Letter 0]

21 Jun 2024

PONE-D-24-20532Mitochondrial DNA is a sensitive surrogate and oxidative stress target in oral cancer cellsPLOS ONE

Dear Dr. Liu,

Thank you for submitting your manuscript to PLOS ONE. After careful consideration, we feel that it has merit but does not fully meet PLOS ONE’s publication criteria as it currently stands. Therefore, we invite you to submit a revised version of the manuscript that addresses the points raised during the review process.

We look forward to receiving your revised manuscript.

Kind regards,

Manisha Nigam

Academic Editor

PLOS ONE

Journal Requirements:

3. Please note that funding information should not appear in the Acknowledgments section or other areas of your manuscript. We will only publish funding information present in the Funding Statement section of the online submission form. Please remove any funding-related text from the manuscript.

4. We note that your Data Availability Statement is currently as follows:

"All relevant data are within the manuscript and its Supporting Information files"

**Additional Editor Comments: **

Based on the opinion of the reviewers and their valuable comments this manuscript cannot be considered for the publication in its present form. It requires major revision on the following points.

Reviewer 1

The authors have presented a well conducted and rigorous research. However the authors should in more or better detail explain why they chose to use normal skin fibroblasts as their control rather than cells that are more similar to the normal oral squamous epithelium for better comparison.

Reviewer 2

The author of the manuscript titled "Mitochondrial DNA is a sensitive surrogate and oxidative stress target in oral cancer cells" claim that mitochondrial DNA of oral cancer cell is sensitive to oxidative stress than normal cells and analysed mtDNA damage, repair and copy number changes by a single method supercoiling-sensitive quantitative PCR (ss-qPCR). However, the experimental strategies presented, lack clarity. The author has tried to prove the potential of method by proving the vulnerability of mtDNA in oral cancer cells against oxidative stress. However, it would be preferable if the author could demonstrate the experimental results obtained from this method compared to other experimental approaches yielding comparable outcomes. The study is not novel.

Reviewers' comments:

Reviewer's Responses to Questions

**Comments to the Author**

1. Is the manuscript technically sound, and do the data support the conclusions?

Reviewer #1: Partly

Reviewer #2: No

2. Has the statistical analysis been performed appropriately and rigorously? 

Reviewer #1: Yes

Reviewer #2: Yes

3. Have the authors made all data underlying the findings in their manuscript fully available?

Reviewer #1: Yes

Reviewer #2: Yes

4. Is the manuscript presented in an intelligible fashion and written in standard English?

Reviewer #1: Yes

Reviewer #2: No

5. Review Comments to the Author

Reviewer #1: The authors have presented a well conducted and rigorous research. However the authors should in more or better detail explain why they chose to use normal skin fibroblasts as their control rather than cells that are more similar to the normal oral squamous epithelium for better comparison.

Reviewer #2: The author of the manuscript titled "Mitochondrial DNA is a sensitive surrogate and oxidative stress target in oral cancer cells" claim that mitochondrial DNA of oral cancer cell is sensitive to oxidative stress than normal cells and analysed mtDNA damage, repair and copy number changes by a single method supercoiling-sensitive quantitative PCR (ss-qPCR). However, the experimental strategies presented, lack clarity. The author has tried to prove the potential of method by proving the vulnerability of mtDNA in oral cancer cells against oxidative stress. However, it would be preferable if the author could demonstrate the experimental results obtained from this method compared to other experimental approaches yielding comparable outcomes. The study is not novel.

6. PLOS authors have the option to publish the peer review history of their article (what does this mean?). If published, this will include your full peer review and any attached files.

Reviewer #1: No

Reviewer #2: No

---

## [Author Response · Author response to Decision Letter 0]

1 Aug 2024

Dear Reviewer,

Thank you for your valuable feedback on our manuscript.We appreciate the time and effort you and the reviewers have dedicated to evaluating our work.We take your feedback seriously and are committed to addressing all points raised.We have carefully considered all your comments, and our replies are outlined as follows.

Journal Requirements:

1.Question: Ensure that your manuscript meets PLOS ONE's style requirements, including those for file naming. 

Answer：The manuscript has been submitted in accordance with the required format.

2.Question：We note that the grant information you provided in the ‘Funding Information’ and ‘Financial Disclosure’ sections do not match. When you resubmit, please ensure that you provide the correct grant numbers for the awards you received for your study in the ‘Funding Information’ section.

Answer：I have revised and proofread the relevant project numbers and names in the 'Funding Information' section. However, I am unable to locate the 'Financial Disclosure' section and cannot make modifications there. I have also uploaded the project numbers as an attachment.

3.Question：Please note that funding information should not appear in the Acknowledgments section or other areas of your manuscript. We will only publish funding information present in the Funding Statement section of the online submission form. Please remove any funding-related text from the manuscript.

Answer:I have removed the acknowledgments section from the manuscript, and there are no references to funding or grants throughout the entire text.

4.Question:We note that your Data Availability Statement is currently as follows:"All relevant data are within the manuscript and its Supporting Information files"Please confirm at this time whether or not your submission contains all raw data required to replicate the results of your study. Authors must share the “minimal data set” for their submission.

Answer:The original data and statistical information corresponding to the images have been uploaded as Supporting Information files, with each folder corresponding to the respective image.

5.Question:PLOS ONE now requires that authors provide the original uncropped and unadjusted images underlying all blot or gel results reported in a submission’s figures or Supporting Information files. 

Answer:The original data images of the gel have been uploaded under the filename "S1_raw_images."

Additional Editor Comments: 

Reviewer 1

The authors have presented a well conducted and rigorous research. However the authors should in more or better detail explain why they chose to use normal skin fibroblasts as their control rather than cells that are more similar to the normal oral squamous epithelium for better comparison.

Answer:First,regarding the choice of cells as controls. In the process of selecting cells for our study, we noted that there were no relevant normal squamous epithelial cell lines available on the ATCC website. Additionally, we found that the experimental designs in relevant literature predominantly utilized normal human dermal fibroblast cell line BJ as a control. For instance, studies such as those by Dominika Szlachcikowska et al. (Int. J. Mol. Sci. 2024, 25, 7329) and Bartosz Skóra et al. (Toxicology and Applied Pharmacology 443 (2022) 116009) have established the use of BJ cells in similar contexts. Therefore, we chose to use normal skin fibroblasts as our control in this study.We hope this clarifies our rationale for the selection of the control cell line.

Reviewer 2

The author of the manuscript titled "Mitochondrial DNA is a sensitive surrogate and oxidative stress target in oral cancer cells" claim that mitochondrial DNA of oral cancer cell is sensitive to oxidative stress than normal cells and analysed mtDNA damage, repair and copy number changes by a single method supercoiling-sensitive quantitative PCR (ss-qPCR). However, the experimental strategies presented, lack clarity. The author has tried to prove the potential of method by proving the vulnerability of mtDNA in oral cancer cells against oxidative stress. However, it would be preferable if the author could demonstrate the experimental results obtained from this method compared to other experimental approaches yielding comparable outcomes. The study is not novel.

Answer:Firstly, during our sensitivity assessment of the experimental methods, previous publications have confirmed that the two-step sensitive quantitative detection method (ss-qPCR) significantly reduces the background levels of mitochondrial DNA damage when compared to conventional PCR methods. This significant reduction in baseline levels is crucial for effectively detecting changes in mitochondrial DNA damage, as traditional methods cannot accurately quantify such damage in mitochondrial DNA.

Additionally, to demonstrate the sensitivity of our method, we conducted flow cytometry and gel electrophoresis experiments. The results confirmed that our ss-qPCR method allows for the detection of damage in both mitochondrial DNA and nuclear DNA. In contrast, flow cytometry and gel electrophoresis primarily assess damage to nuclear DNA and are not effective for detecting mitochondrial DNA damage.

We appreciate your understanding and hope this adequately addresses your concerns.

Sincerely,

Haiwen Liu

---

## [Editor Report · Decision Letter 1]

15 Aug 2024

Mitochondrial DNA is a sensitive surrogate and oxidative stress target in oral cancer cells

PONE-D-24-20532R1

Dear Dr. Liu,

We’re pleased to inform you that your manuscript has been judged scientifically suitable for publication and will be formally accepted for publication once it meets all outstanding technical requirements.

Kind regards,

Manisha Nigam

Academic Editor

PLOS ONE

---

## [Editor Report · Acceptance letter]

21 Aug 2024

PONE-D-24-20532R1 

PLOS ONE

Dear Dr. Liu, 

I'm pleased to inform you that your manuscript has been deemed suitable for publication in PLOS ONE. Congratulations! Your manuscript is now being handed over to our production team.

Kind regards, 

on behalf of

Dr. Manisha Nigam 

Academic Editor

PLOS ONE